

# Refining amino acid hydrophobicity for dynamics simulation of membrane proteins

Ronald D. Hills, Jr

Department of Pharmaceutical Sciences, College of Pharmacy, University of New England, Portland, ME, United States of America

## ABSTRACT

Coarse-grained (CG) models have been successful in simulating the chemical properties of lipid bilayers, but accurate treatment of membrane proteins and lipid-protein molecular interactions remains a challenge. The CgProt force field, original developed with the multiscale coarse graining method, is assessed by comparing the potentials of mean force for sidechain insertion in a DOPC bilayer to results reported for atomistic molecular dynamics simulations. Reassignment of select CG sidechain sites from the apolar to polar site type was found to improve the attractive interfacial behavior of tyrosine, phenylalanine and asparagine as well as charged lysine and arginine residues. The solvation energy at membrane depths of 0, 1.3 and 1.7 nm correlates with experimental partition coefficients in aqueous mixtures of cyclohexane, octanol and POPC, respectively, for sidechain analogs and Wimley-White peptides. These experimental values serve as important anchor points in choosing between alternate CG models based on their observed permeation profiles, particularly for Arg, Lys and Gln residues where the all-atom OPLS solvation energy does not agree well with experiment. Available partitioning data was also used to reparameterize the representation of the peptide backbone, which needed to be made less attractive for the bilayer hydrophobic core region. The newly developed force field, CgProt 2.4, correctly predicts the global energy minimum in the potentials of mean force for insertion of the uncharged membrane-associated peptides LS3 and WALP23. CgProt will find application in studies of lipid-protein interactions and the conformational properties of diverse membrane protein systems.

# INTRODUCTION

Coarse-grained (CG) simulations greatly expand the timescale of events (*Kmiecik et al., 2016*; *Venable et al., 2017*) that can be studied in physiologically realistic membrane systems (*Ma et al., 2015*; *Van Oosten & Harroun, 2016*). Reducing the vast number of atomic degrees of freedom in the many membrane lipids, large transmembrane protein constituents, and the surrounding bulk water is a necessary step that requires an accurate and transferable set of interaction potentials in the reduced conformational space (*Bereau, Wang & Deserno, 2014*). Construction of a standalone force field for the twenty amino

Corresponding author
Ronald D. Hills, Jr, rhills@une.edu

acids and common membrane lipids enables molecular dynamics (MD) simulation of proteins of arbitrary sequence, structure and size (*Bereau, Wang & Deserno, 2014*; *Ganesan & Matysiak, 2014*; *Han, Wan & Wu, 2008*). By enabling membrane proteins to be studied in the context of their native environment, CG simulations have yielded numerous insights into the nature and functional role of the dynamical interplay between proteins and the lipid bilayer (*Bennett & Tieleman, 2013*; *Hedger & Sansom, 2016*; *Marrink & Tieleman, 2013*; *Poyry & Vattulainen, 2016*).

A natural method for assessing the balance of energetics in a CG model is to use umbrella sampling simulations to calculate the potential of mean force (PMF) for dragging sidechain analog compounds through the bilayer (*MacCallum, Bennett & Tieleman, 2008*). Agreement with the corresponding results from atomistic MD simulation has been used to assess a variety of CG representations such as Martini (*De Jong et al., 2013*), PRIMO (*Kar et al., 2014*) HMMM (*Pogorelov et al., 2014*), ELBA (*Genheden & Essex, 2015*) and others (*Vorobyov et al., 2016*). The PMFs generated from atomistic MD of sidechains such as polar glutamine and basic arginine/lysine, however, have been observed to deviate from various experimental measures of solvation energy. Moreover, such studies neglect the influence of both the polypeptide backbone and the occlusion of solvent-accessible surface area by neighboring amino acid residues in a protein chain (*Singh & Tieleman, 2011*; *Wimley, Creamer & White, 1996*).

Experimental measures for the hydrophobicity of isolated sidechains involve the partitioning of analog compounds (methane for Ala, propane for Val, etc.) in cyclohexane-water and octanol-water mixtures (*Radzicka & Wolfenden, 1988*). To estimate the solvation energy for residues occluded by moderate-sized neighbor residues in the peptide chain, the octanol-to-water partitioning of Ac-WL-X-LL pentapeptides was determined (*Wimley, Creamer & White, 1996*) relative to AcWL-G-LL (see $\Delta G^{cor}$ in Table 2 of Wimley et al.). Combining these values with the estimated 1.15 kcal/mol backbone penalty for nonpolar solvation of the glycyl -$CH_2CONH$- unit yielded a whole-residue hydrophobicity scale that is useful for predicting transmembrane regions in proteins (*White & Wimley, 1998*). For a measure of exposed sidechains, solvation energies were corrected for occlusion by the host peptide using measurements of the nonpolar solvent-accessible surface area as compared to Ac-GG-X-GG (see $\Delta G^{GXG}$ in Table 2 of Wimley et al.). Another measure for the fully exposed residue is the octanol-water solvation energy of an acetyl amino acid amide dipeptide: Ac-X-amide (*Fauchere & Pliska, 1983*). Experimental values are compiled in the Supplemental Information (Table S1, CSV format).

In the present work, parameter modifications of the CgProt force field (*Hills Jr, Lu & Voth, 2010*; *Ward, Guvench & Hills Jr, 2012*) are explored by comparing their resulting sidechain permeation profiles in a DOPC bilayer. By reassigning the CG site type definitions for select sidechain interaction centers, the permeation profiles of several amino acids are found to improve relative to either atomistic simulation results or the known experimental solvation energies. The sidechain parameters are then coupled to an improved description of the peptide backbone. The resulting force field, CgProt version 2.4, is assessed for the ability of designed helical peptides to remain inserted in the membrane during microsecond MD simulations. While the previous version, CgProt 2.3, relied on terminal charges to anchor

transmembrane peptides across the bilayer, the current force field captures the proper orientation of neutral peptides LS3 and WALP23 in the bilayer, consistent with experiments using capped uncharged termini. The refinement of amino acid hydrophobicity makes CgProt a powerful tool for studying lipid-protein interactions.

## METHODS

### Coarse grain model

The first iteration of the CgProt force field was developed for aqueous proteins using the multiscale coarse graining (MS-CG) method (*Hills Jr, Lu & Voth, 2010*). MS-CG is a variational force-matching procedure for developing a self-consistent set of CG interaction potentials given a reference ensemble generated from atomistic MD (*Noid et al., 2008*). A strength of MS-CG is that it can map an inherently many-body PMF into effective pairwise interactions without an *a priori* assumed functional form (*Noid et al., 2007*). To construct a protein force field, tabulated nonbond potentials were developed for five unique CG site types *ca/p/ap/pos/neg* assigned to groups of backbone and sidechain atoms based on atomistic reference simulations of peptide unfolding in water. Because the model detail emphasized the amino acid sidechains rather than backbone, CgProt itself is not useful for *de novo* folding of proteins. When deployed with an elastic network encoding the backbone native conformation, the force field can effectively simulate protein conformational transitions (*Ward, Guvench & Hills Jr, 2012*).

To enable the simulation of membrane proteins, atomistic simulations were performed of peptide unfolding in aqueous mixtures of phospholipids. MS-CG was then used to develop lipid-protein interaction potentials for the headgroup and tail sites: *ch/hh/ph/gl/e1/e2/s1/sd/sm* for common physiological lipids (*Hills Jr & McGlinchey, 2016*; *Ward, Guvench & Hills Jr, 2012*). For the lipid-lipid interactions needed to construct a membrane model, parameters were adapted from a previous MS-CG model developed for a DOPC:DOPE bilayer in the absence of explicit water molecules (*Lu & Voth, 2009*). The present work assesses the physical realism that can be obtained by combining the two protein and lipid force fields.

The polypeptide is represented as a chain of backbone beads defined at the alpha carbon positions with up to four CG sites defined within each sidechain as the mass centers of non-overlapping subgroups of atoms. In contrast to approximations made in other CG models in order to obtain larger time steps, Boltzmann inversion is used to assign accurate harmonic interaction potentials that reproduce the CG bond fluctuations observed in atomistic MD (*Hills Jr, 2014*). Fourth order polynomials were used to fit the 3-body angle distributions observed for each of the 20 amino acids (*Hills Jr, Lu & Voth, 2010*). The Cα backbone is assigned generic but sequence-dependent torsional potentials that allow for the adoption of either α-helix or β-sheet secondary structure (*Karanicolas & Brooks III, 2002*). The original CgProt parameterization employed a unique site type for the backbone Cα bead: *ca*. To reproduce the bilayer permeation profiles of backbone moieties, version 2.4 assigns the existing apolar site type (ap) to Ala residues and the existing polar site (p) type to all other backbone beads (Fig. 1). New site type assignments are explored for select

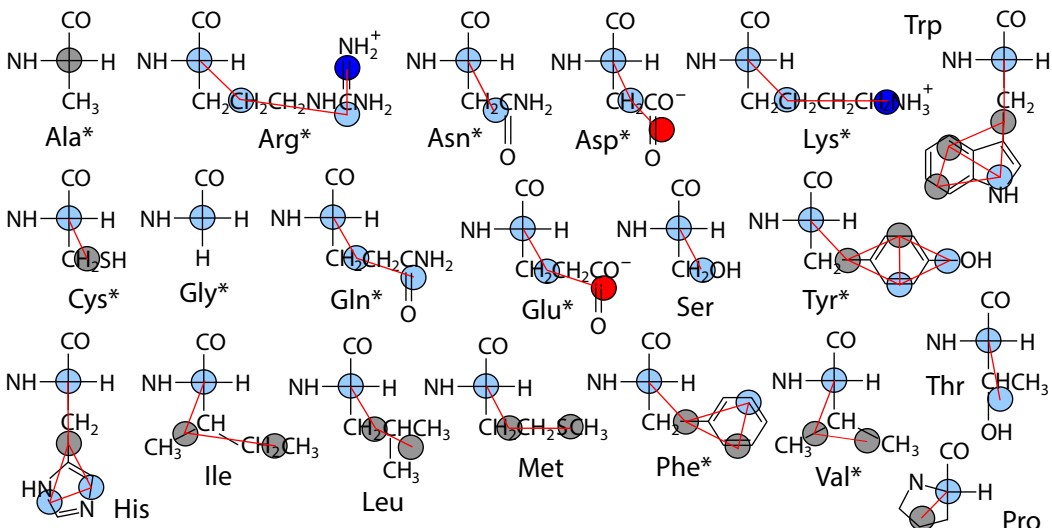

**Figure 1 Site assignments in CgProt 2.4.** Alpha carbons not involving alanine are reassigned the polar site type (sky blue). Other nonbond interaction types include positive (blue), negative (red), and apolar (gray). Asterisks denote new sidechain descriptions for version 2.4.

sidechain $p$ or $ap$ sites, but the original nonbond potentials tabulated for each site type pair combination were not modified.

Given the lack of explicit water, dynamics employ the stochastic Langevin dynamics routine with a damping friction coefficient of $\gamma = 0.5$ ps$^{-1}$, a significant speedup relative to protein dynamics in real water (*Cerutti et al., 2008*). The lateral diffusion of lipids in the bilayer is enhanced 30-fold in CgProt (*Fosso-Tande et al., 2017*). An effective temperature of $kT = 2.24$ kJ/mol is used in the present work as it has been shown to reproduce lipid bilayer structure across different liquid phase systems (*Hills Jr & McGlinchey, 2016*). Nonbond interactions have a 1.2 nm cutoff and the neighbor list is updated every step. As recommended by others (*Arnarez et al., 2015*), the implicit solvent bilayer simulations were performed in the fixed volume NVT ensemble. The Gromacs 4.6.7 simulation package was used (*Pall et al., 2015*) with a 5 fs integration time step.

## Sidechain insertion PMFs

CgProt is assessed and refined by comparing the energetics of sidechain bilayer insertion to PMFs from atomistic simulations reported in the literature (*De Jong et al., 2013*; *Johansson & Lindahl, 2008*; *MacCallum, Bennett & Tieleman, 2008*). Umbrella sampling calculations were performed with a harmonic force constant of 350 kJ mol$^{-1}$ nm$^{-2}$ applied between the mass centers of the sidechain and lipid bilayer in the $z$-dimension. As in previous atomistic studies (*MacCallum, Bennett & Tieleman, 2008*), an additional copy of the sidechain was placed 3.7 nm above the first in order to duplicate the number of data collected in a single run. A total of 21 umbrella windows with 0.2 nm spacing were simulated for 50 ns each. To ensure overlap between successive windows, sidechain 1 was restrained at depths spanning $z = -3.3$ to $z = +0.5$ nm from the bilayer center, while sidechain 2 spanned $z = 0$ to 4
nm. The two halves of the PMFs were computed using g_wham (*Hub, De Groot & Van der Spoel, 2013*) in Gromacs and treated as independent samples to determine the largest error over the 400 bins. After shifting to zero in bulk water, the maximum PMF error for most sidechains was between 0.1 and 0.5 kJ/mol at the bilayer center ($z = 0$). Exceptions were Arg, Lys and Asp, whose strong repulsion at $z = 0$ had a maximum error of 2.0–2.6 kJ even after increasing the simulation length to 250 ns.

A large periodic simulation box (256 DOPC molecules, 9.4 nm box edge) was used to minimize boundary effects in the umbrella simulations (*Neale & Pomes, 2016*; *Nitschke, Atkovska & Hub, 2016*). By comparison, a 250 ns simulation performed with a small box (64 DOPC, $4.7 \times 4.7 \times 9$ nm) containing two arginine sidechains reveals that the small simulation significantly overestimates the steepest part of the PMF at a distance of $z = 1$ nm from the bilayer center. The larger box also decreases the likelihood that the two sidechain copies influence each other. A comparison PMF obtained from a simulation with only one arginine sidechain in a 9.4 nm box fell within the 2.6 kJ margin of error of the two sidechain simulation.

**Peptide insertion simulations**

The membrane binding or insertion behavior of synthetic peptides in the DOPC bilayer was used to test the balance of membrane-protein interactions in CgProt, as has been performed with other force fields (*Bereau et al., 2015*; *Bereau & Kremer, 2016*; *Bereau, Wang & Deserno, 2014*; *Bond et al., 2007*; *Hall, Chetwynd & Sansom, 2011*; *Kar et al., 2014*; *Pulawski et al., 2016*; *Ward, Nangia & May, 2017*). The sequence GWW(LA)$_8$LWWA (WALP23) of the WALP $n$ series of single-pass transmembrane peptides was simulated since its 25.5 Å hydrophobic length (*Kim & Im, 2010*) nearly matches the hydrophobic thickness of the DOPC bilayer: 29 Å as measured by the mean distance between ester sites. The highly charged transmembrane peptides $K_2L_{24}K_2$ and $K_2(LA)_{12}K_2$, referred to as $L_{24}$ and $(LA)_{12}$, respectively, were also assessed for their bilayer insertion stability (*Liu et al., 2004*; *Zhang et al., 1995*). Last, the designed amphipathic peptide (LSSLLSL)$_3$, referred to as LS3, was tested for its binding at the membrane-water interface (*Lear, Wasserman & DeGrado, 1988*).

In versions 2.3 and earlier CgProt relied on terminal charges to anchor peptides within the membrane (*Fosso-Tande et al., 2017*). In this work, each N- and C-terminus is kept neutral in order to rigorously assess the insertion behavior of the current force field and its new backbone description. This is consistent with experimental studies (*Holt et al., 2009*; *Zhang et al., 1995*) of the designed peptides in which the N- and C-termini are capped with acetyl and amide groups, respectively. Each of eight simulation runs were conducted for 2 µs in a $9.5 \times 9.5 \times 10$ nm box containing 256 DOPC molecules, with the peptide starting in either a membrane-bound or completely desorbed state. Peptides were restrained in an ideal right-handed α–helix conformation by applying a harmonic GROMOS-96 angle ($\theta_0 = 89°$, $k_\theta = 2{,}000 \, \text{kJ mol}^{-1}$) between each successive Cα-Cα-Cα and a harmonic torsion ($\varphi_0 = 50°$, $k_\varphi = 2{,}000 \, \text{kJ mol}^{-1} \, \text{rad}^{-2}$) for Cα-Cα-C α-Cα terms. The angle of peptide insertion with respect to the bilayer normal was calculated using the vector spanning alpha carbons 5 and 16 for LS3, 5 and 19 for WALP23, and 8 and 26 for $L_{24}/(LA)_{12}$.

### Peptide insertion PMFs

The PMFs for the insertion of LS3 and WALP23 peptides in a DOPC bilayer were determined using umbrella sampling simulations (*Bereau et al., 2015*; *Bereau & Kremer, 2016*; *Ward, Nangia & May, 2017*). Umbrella windows incorporated a single peptide placed at 0.2 nm intervals along the $z$-axis. Starting structures were generated by pulling the peptide into the bilayer over a 4.6 ns simulation ($2 \times 10^4$ kJ mol$^{-1}$ nm$^{-2}$ force constant) starting from either side of the bilayer at $z = \pm 4.6$ nm. For each peptide, a total of forty-eight windows were simulated for 100 ns each using a force constant of 650 kJ mol$^{-1}$ nm$^{-2}$ force constant for the peptide center of mass relative to the bilayer normal. Snapshots were recorded every 1 ps. The first 16 ns of simulation time was excluded from the WHAM calculation to allow for the peptide orientation to equilibrate.

## RESULTS AND DISCUSSION

### Sidechain permeation profiles

The DOPC bilayer can be divided into four regions (Fig. 2) to aid in comparing to previous work (*Genheden & Essex, 2015*; *MacCallum, Bennett & Tieleman, 2008*). The tail region, found within 1 nm of the bilayer center, contains only hydrophobic tails. The ester region from $z = 1$ to 1.8 nm begins at the minimum depth of the carbonyls, has a falling lipid tail density, and contains most of the glycerol density. The head group region from 1.8 to 2.8 nm contains the largest portion of the phosphate and choline head group density. The fourth region represents the pure water phase. These trends in DOPC lipid partial densities are in agreement with atomistic MD simulations (*MacCallum, Bennett & Tieleman, 2008*).

PMF insertion profiles are reported for each sidechain from $z = 0$ to 3.7 nm (Fig. 3). For sidechains in the original force field that did not exhibit expected behavior, alternate CG descriptions were explored. Note that Cys and Pro sidechains consisting of a lone apolar bead are indistinguishable in absence of the C$\alpha$ site, as are polar Ser and Thr. The C$\beta$-C$\gamma$ virtual bond between apolar CG sites is nearly identical in Ile and Met and leads to PMF profiles that are indistinguishable. The single polar site for Ser/Thr has a 4.4 kJ/mol attractive well at $z = 1.7$ nm and a 10 kJ barrier to crossing the bilayer center at $z = 0$. This compares well to the atomistic PMFs obtained in OPLS for Ser and Thr, which have 1–4 kJ attraction at $z = 1.5$ nm and a repulsive barrier for $z < 1$ nm that reaches 15 kJ (*MacCallum, Bennett & Tieleman, 2008*). For Cys, the single apolar site has an 8.5 kJ repulsion at $z = 2.1$ nm and is attractive by 4–6.6 kJ for $z < 1$ nm. This corresponds well with the atomistic PMF reported by MacCallum et al. for cysteine: 3 kJ repulsion at $z = 2.1$ nm and 4–7 kJ attraction for $z < 1.5$ nm. In contrast, the orginal CgProt force field (v.1) employed a polar site for cysteine, overestimating the energy of insertion into the tail region by 13 kJ/mol. For Val, a second apolar site was added to its CG description to capture the hydrophobic nature of the atomistic PMF for valine: 5 kJ repulsion at $z = 2$ nm and >10 kJ attraction for $z < 1.5$ nm.

Ile and Leu also contain two apolar sites, having a 17–22 kJ attraction for the tail region of the bilayer. The short 1.6 Å C$\beta$–C$\gamma$ bond of Leu and Val allows them to pack more favorably inside the tail region than Met and Ile (2.5 Å bond). The PMFs are in qualitative
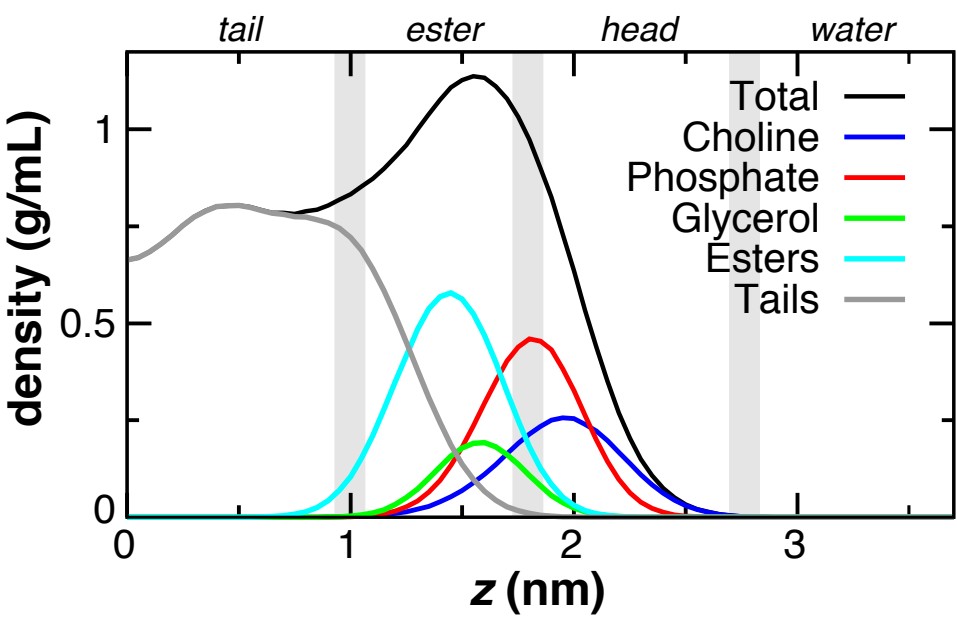

**Figure 2** **Partial lipid density profiles relative to the bilayer center ($z = 0$) for the CG DOPC simulation.** Vertical gray bars divide the bilayer into four regions: hydrophobic tails, esters, lipid head groups, and solvent.

agreement with the 15–22 kJ attraction for $z < 1$ nm in the Ile and Leu atomistic profiles (*MacCallum, Bennett & Tieleman, 2008*), but there is a larger repulsion in the head group region. The three apolar sites in the original description for Phe overestimate its atomistic PMF, which bears a 15 kJ attraction for $z < 1.5$ nm (*De Jong et al., 2013*; *MacCallum, Bennett & Tieleman, 2008*). Better agreement for aromatic phenylalanine was found by assigning the polar site type to C$\delta$. Neutral histidine contains one apolar and two polar sites in CgProt with a 14 kJ attraction at $z = 1.5$ nm and 8 kJ repulsion at $z = 0$. The atomistic PMF for His is similar in these features but has a smaller energy barrier in the vicinity of $z = 2.5$ nm (*De Jong et al., 2013*). Using three polar sites for His reduces the barrier but results in too strong an attraction for the head group region.

The atomistic PMFs for Trp and Tyr are 21 and 13 kJ attractive at $z = 1.3$ nm and rise to $-5$ and $+7$ kJ at $z = 0$, respectively (*MacCallum, Bennett & Tieleman, 2008*). The best agreement for tryptophan in CgProt is to assign three apolar sidechain sites along with one polar site at C$\gamma$. For tyrosine, assignment of polar sites at C$\gamma$ and C$\varepsilon$ in the current version improves the permeation profile in the bilayer tail region. A significant energy barrier is found in both potentials at $z = 2.5$ nm. As this falls outside the lipid density of the bilayer (Fig. 2), the barrier is likely to affect the kinetics of insertion but not the thermodynamic stabilization for proteins residing in the membrane.

The atomistic PMFs for Gln and Asn are 9 and 7 kJ attractive at $z = 1.5$ nm and 20 and 24 kJ repulsive at $z = 0$, respectively (*De Jong et al., 2013*; *MacCallum, Bennett & Tieleman, 2008*). To reproduce this behavior, the C$\beta$ site type of glutamine was changed from apolar to polar, resulting in stronger attraction at the membrane interface and stronger repulsion in

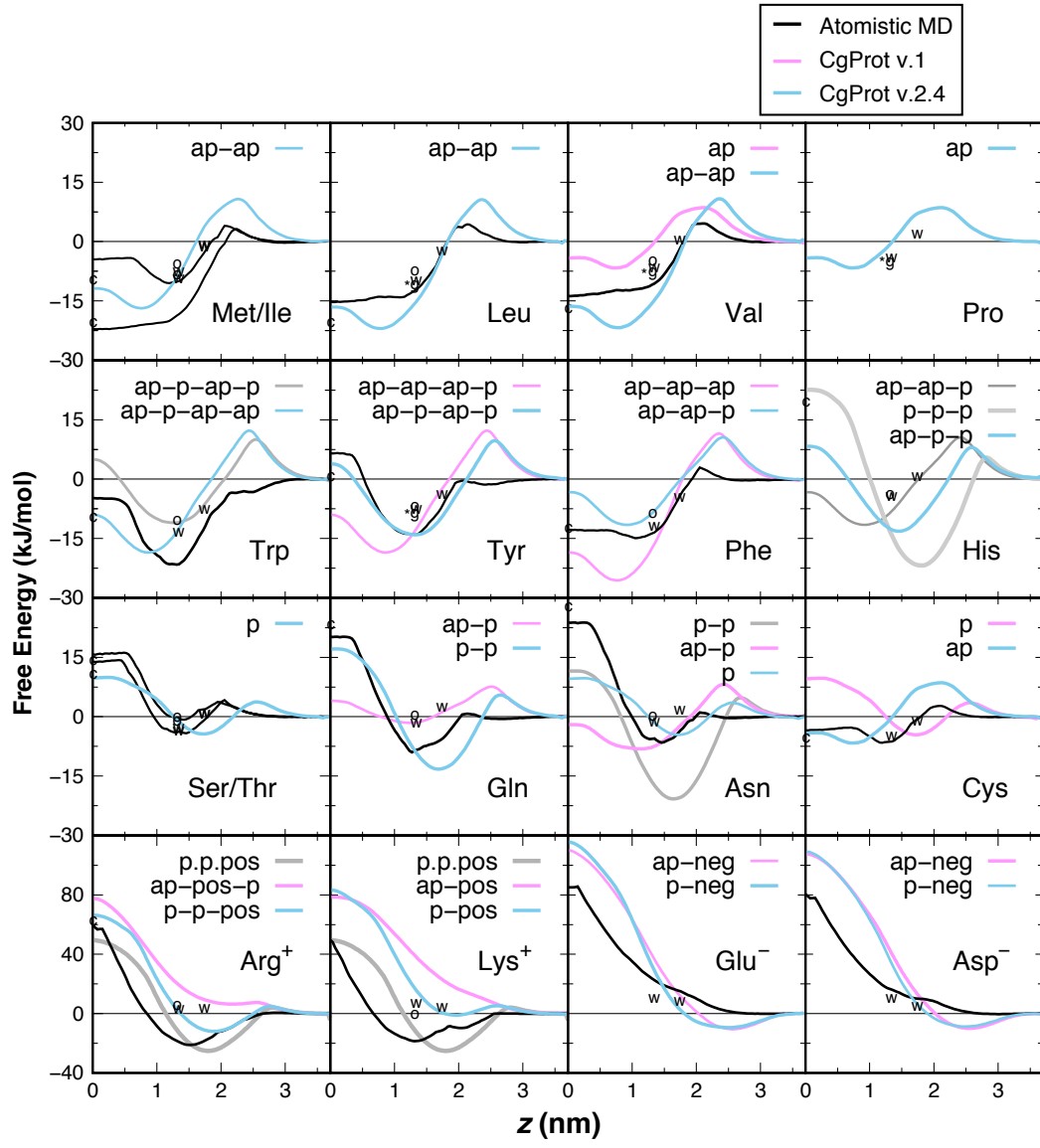

**Figure 3** **PMF insertion profiles for mass centers of sidechain analogs relative to the bilayer center.**
PMFs are set to zero in water. PMFs from atomistic simulations are compared in black (*MacCallum, Bennett & Tieleman, 2008*). The bonding of CG sites is depicted for polar (p), apolar (ap), positive (pos) and negative (neg) types. Shorter bonds (.) were tested for Lys/Arg. Alternate sidechain descriptions shown in gray are not recommended. Various experimental data are plotted relative to zero for glycine. For sidechain analogs, the water-cyclohexane transfer energy is marked $c$ at $z = 0$ and the water-octanol energy is marked $o$ at $z = 1.3$ nm (*Radzicka & Wolfenden, 1988*). $\Delta G^{cor}$ for the octanol solvation of a sidechain in a peptide occluded by moderate-sized neighbors is marked $w$ at $z = 1.3$ nm (*Wimley, Creamer & White, 1996*). $\Delta G^{GXG}$ for a fully exposed sidechain in GGXGG peptide is marked *g at 1.3 nm for Leu, Val, Pro and Tyr. The sidechain solvation energy for a WLXLL peptide at the POPC interface is marked $w$ at $z = 1.7$ nm (*Wimley & White, 1996*).

the hydrophobic tail region. For asparagine, the best agreement was obtained by removing its apolar site altogether, which involved increasing the Cα–Cβ bond from 2.0 to 2.6 Å. Glu and Asp are the only sidechains with a monotonic repulsion observed in atomistic MD for all regions in the bilayer, along with an 80 kJ barrier to crossing the bilayer in their charged form. CgProt reproduces the strong repulsion in the tail and ester regions using one polar and one negative site type. In place of a screened Coulomb potential, the force field energy function successfully combines electrostatic and van der Waals forces into a single tabulated function for each of the positive and negative site types (*Lu & Voth, 2009*; *Noid et al., 2008*).

The atomistic PMFs for Arg$^+$ and Lys$^+$ are +50 kJ repulsive at $z = 0$ but have a large attractive well spanning $z = 0.8$ to 2.5 nm with a minimum of $-20$ kJ/mol (*De Jong et al., 2013*; *MacCallum, Bennett & Tieleman, 2008*). Arginine and lysine were too repulsive for the membrane in CgProt v.1. The solution employed in CgProt 2.4 is to reassign their apolar Cβ sites to be polar. An alternate sidechain description was also tested consisting of two polar sites and a positive site connected by short 1.5 Å bonds (p.p.pos). Experimental measures of hydrophobicity were used to discriminate between alternate CG descriptions, which are discussed in the next section.

There has been much interest in the attraction of positive-charged sidechains for the lipid bilayer (*Gumbart & Roux, 2012*; *MacCallum & Tieleman, 2011*; *Sun, Forsman & Woodward, 2015*). Findings are now emerging as to its potential functional relevance for peptides (*Gleason et al., 2013*; *Hu, Sinha & Patel, 2014*; *MacCallum, Bennett & Tieleman, 2011*; *Nakao et al., 2016*; *Rice & Wereszczynski, 2017*). The favorable attraction of basic residues for the ester bilayer region has been dubbed snorkeling. A bilayer defect is created in which polar lipid head groups dive down to solvate the positive charge. A representative snapshot from the CgProt umbrella simulations (Fig. 4) shows the arginine sidechain near the center of the bilayer producing local changes in the depth of glycerol, phosphate and choline groups, similar to results observed in atomistic MD (*MacCallum, Bennett & Tieleman, 2008*).

## Experimental partition energies

Experimental values for the partitioning of sidechain analog compounds from water to cyclohexane were correlated with the free energy of transfer to the bilayer center at $z = 0$ (*MacCallum, Bennett & Tieleman, 2008*; *Pogorelov et al., 2014*; *Vorobyov et al., 2016*) computed from the CgProt 2.4 PMFs. A linear regression was performed for all sidechains except charged Asp, Glu and Lys. Since the experiments did not control the ionization state (*Radzicka & Wolfenden, 1988*), the Asp, Glu and Lys sidechains are expected to be neutral in the nonpolar environment of cyclohexane based on analysis of their pKa values and solvation energies from atomistic MD (*MacCallum, Bennett & Tieleman, 2008*). A strong correlation (Pearson coefficient of $r = 0.95$) is observed between CgProt 2.4 and experiment (Fig. 5A), similar to the correlation between cyclohexane solvation energies and the atomistic PMF values of MacCallum et al. (Table 1).

The transfer free energy of sidechain analogs from water to octanol was measured in the same study (*Radzicka & Wolfenden, 1988*). Positing that octanol has properties analogous

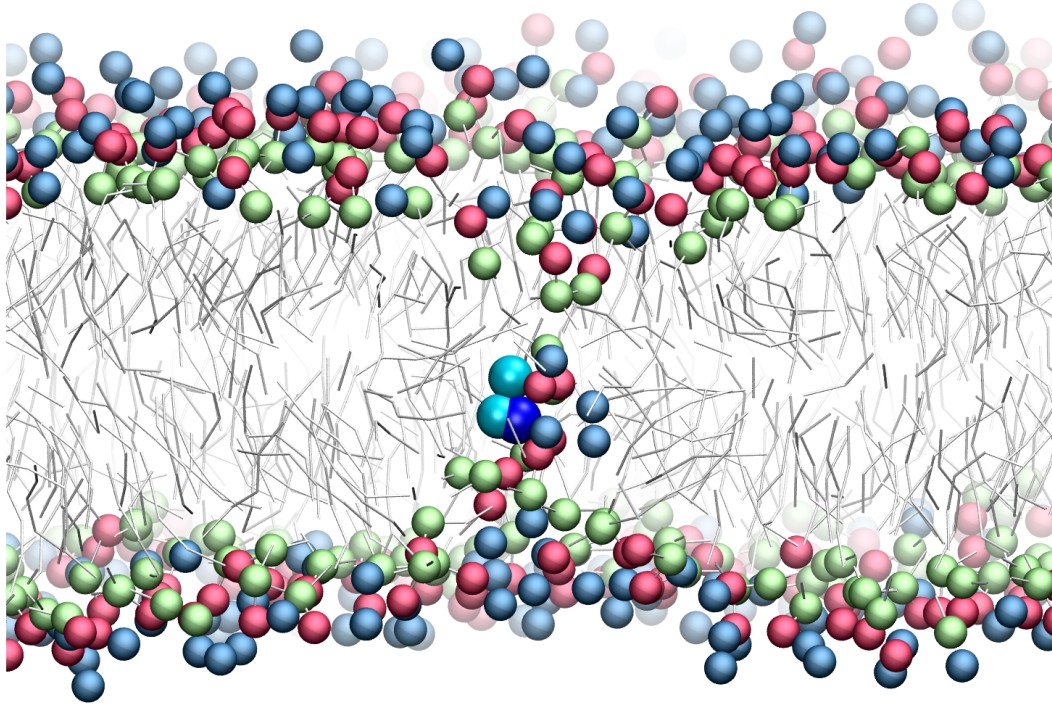

**Figure 4** **Snapshot of p-p-pos Arg (cyan-cyan-blue spheres) restrained near the center of the bilayer.** A bilayer defect solvates the CgProt sidechain similar to observations in atomistic MD (*MacCallum, Bennett & Tieleman, 2008*). DOPC head groups are colored as blue (choline), red (phosphate) and green (glycerol) balls. Lipid tails are traced as sticks.

**Table 1** **MacCallum et al. and CG PMFs versus experiment.**

| *x:* Experiment ($\Delta G$) | *y:* PMF | Pearson correlation, *r* | Slope | Intercept (kJ/mol) |
|---|---|---|---|---|
| Cyclohexane-water | Atomistic[a] | 0.98 | 0.89 | 1.3 |
| Cyclohexane-water | CgProt[a] | 0.95 | 0.88 | −1.2 |
| Octanol-water | Atomistic[b] | 0.92[c] | 1.6 | −2.6 |
| Octanol-water | CgProt[b] | 0.78 | 1.1 | −3.7 |
| W-W octanol ($\Delta G^{cor}$) | Atomistic[b] | 0.95[c] | 1.6 | 0.8 |
| W-W octanol ($\Delta G^{cor}$) | CgProt[b] | 0.91 | 2.0 | 5.3 |
| W-W POPC interface | Atomistic[d] | 0.86[c] | 1.6 | −1.6 |
| W-W POPC interface | CgProt[d] | 0.51[e] | 0.68 | −0.4 |

**Notes.**
[a] $z = 0$ nm.
[b] $z = 1.3$ nm.
[c] Excludes outliers Arg and Lys.
[d] $z = 1.7$ nm.
[e] Excludes outliers His, Arg and Gln.

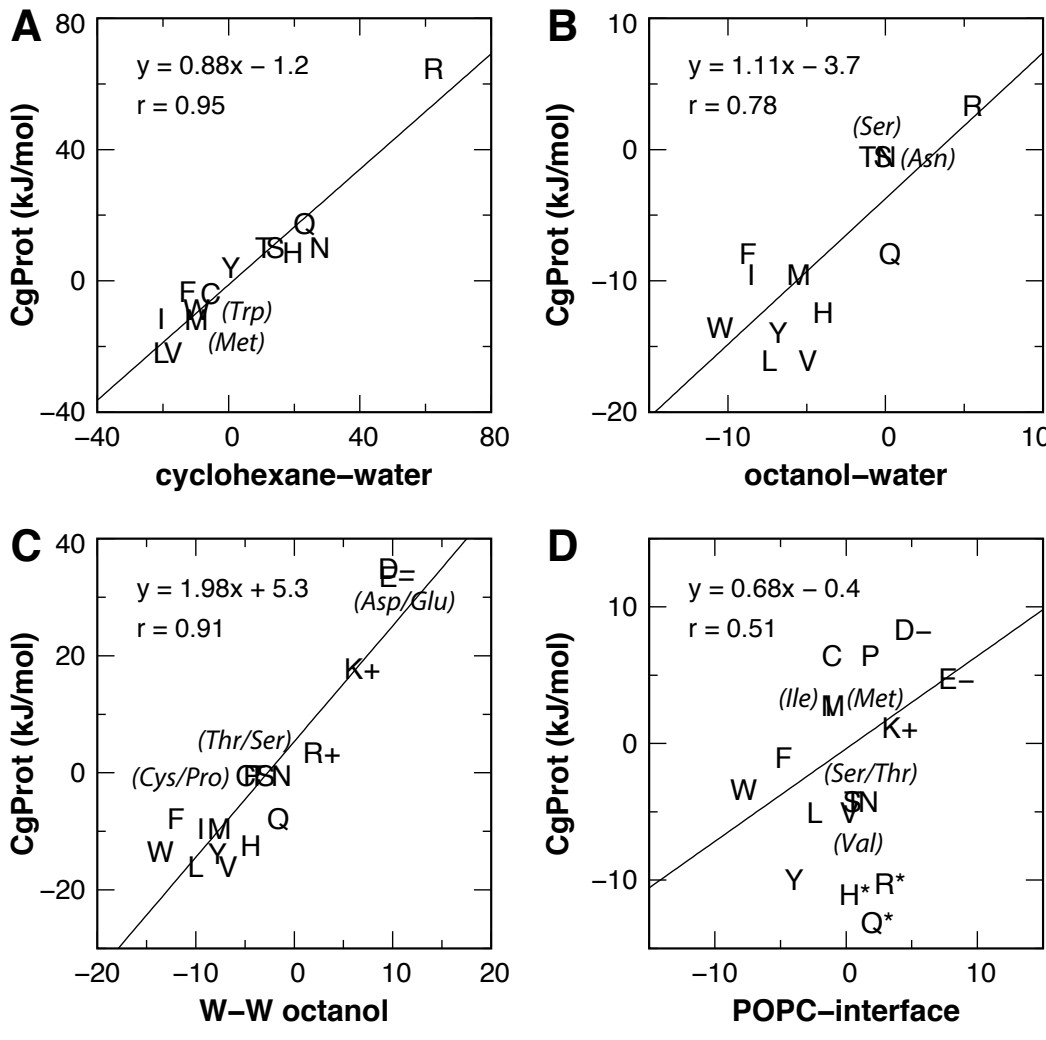

**Figure 5** **Linear regression of CgProt 2.4 sidechain solvation energies with experimental data.** Single letter amino acid codes are plotted along with select 3-letter codes in parentheses. Water to cyclohexane (A) and water to octanol (B) transfer energies of unionized sidechain analogs (*Radzicka & Wolfenden, 1988*) are correlated to PMF values at $z = 0$ and $z = 1.3$ nm, respectively. Wimley-White peptide residue transfer energies relative to glycine are correlated to PMF values at $z = 1.3$ and $1.7$ nm for octanol-water (C) and the POPC interface (D), respectively.

to the ester region of the bilayer, we correlated the octanol solvation energies to the PMF values of MacCallum et al. for transferring solute to a membrane depth of $z = 1.3$ nm and obtained a good correlation of $r = 0.92$ upon exclusion of outliers Arg and Lys (Table 1). For the PMFs calculated in CgProt 2.4, a correlation of $r = 0.78$ was found upon exclusion of sidechains Asp, Glu and Lys (Fig. 5B). Arginine, which is predicted to be protonated in the bilayer ester region (*MacCallum, Bennett & Tieleman, 2008*), falls on the regression line.

For a measure of sidechains occluded by neighboring residues of moderate size (*Wimley, Creamer & White, 1996*), the octanol-water partitioning of WLXLL peptides relative to glycine ($\Delta G^{\text{cor}}$) was correlated with the PMF values at $z = 1.3$ nm. A strong correlation ($r = 0.95$) is observed for the atomistic PMFs, excluding outliers Arg and Lys. Similarly, a prominent linear correlation ($r = 0.91$) is observed for all amino acid sidechains in CgProt (Fig. 5C). The experiments of Wimley et al. controlled for protonation state, allowing for comparison to all charged sidechain simulations. The CgProt PMFs also correlate well with $\Delta G^{\text{GXG}}$ values for a fully exposed residue in a glycine peptide ($r = 0.91$).

To compare the partitioning of sidechains to the polar environment of the bilayer interface, we take the whole-residue solvation energies derived from the WLXLL peptide series in POPC and subtract the $+0.04$ kJ/mol value for glycine (*Wimley & White, 1996*). Using the free energy at a depth of $z = 1.7$ nm, a good correlation of $r = 0.86$ is obtained for the atomistic PMFs of MacCallum et al., excluding outliers Arg and Lys. For CgProt, a weak correlation of $r = 0.51$ is observed. His, Arg and Gln were found to be outliers and were excluded from the linear regression (Fig. 5D). The various transfer energies for cyclohexane, octanol, and POPC measurements are plotted at $z = 0$, 1.3, and 1.7 nm, respectively, in Fig. 3. Together the experimental data points serve as a guide for selecting between alternate CG descriptions. This is particularly useful for Arg and Lys, for which more positive solvation energies are measured in experiments as compared to atomistic MD.

## Peptide backbone parameters

Earlier versions of CgProt enabled the rapid insertion of peptides with charged N- and C-termini in unbiased MD simulations (*Fosso-Tande et al., 2017*). This was due to the attractive nature of the unique backbone type (*ca*) for the bilayer core. PMFs are calculated for the insertion of one and two-residue ca backbone units in the bilayer, revealing a free energy minimum at $z = 0.7$ nm of $-16.5$ and $-35$ kJ/mol, respectively (Fig. 6). This is in stark contrast with the $+1.15$ kcal/mol water to octanol transfer energy for the glycyl -CH$_2$CONH- unit (*Wimley, Creamer & White, 1996*). PMFs calculated for one and two-residue backbone units using the polar site type (*p*) are in better agreement with PMFs obtained from atomistic MD for acetamide and other backbone mimics (*Bereau & Kremer, 2016*; *Sandoval-Perez, Pluhackova & Bockmann, 2017*), as well as the positive partition energies obtained from experiment. For the CgProt representation of alanine, the backbone bead is the sole CG site. The PMF of methane (CH$_4$) has a $-8.4$ kJ/mol free energy minimum at $z = 0$ and its cyclohexane partition energy is $-7.6$ kJ/mol (*Radzicka & Wolfenden, 1988*). Alanine residues are therefore represented using the apolar site type (*ap*), which provides good agreement with the Wimley-White whole-residue values (Fig. 6).

## Peptide insertion properties

Simulations of peptide insertion were used to test the balance of backbone and sidechain interactions in CgProt 2.4 (*Bereau & Kremer, 2016*). Multiple independent simulations were performed for the designed helical peptides LS3, WALP23, L$_{24}$ and (LA)$_{12}$ starting from either membrane-bound or desorbed states. In each simulation, the peptide adopted a

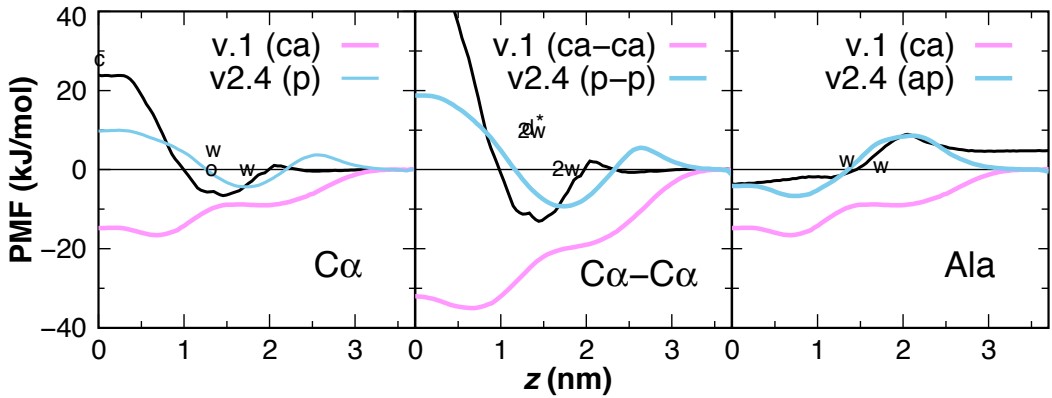

**Figure 6** **CgProt PMFs for one and two-residue backbone moieties.** CgProt v.1 employed a unique alpha carbon site type (ca) as opposed to polar sites (p). Cα: The atomistic PMF for acetamide is drawn in black as an approximation for the glycyl -CH$_2$CONH- unit (*MacCallum, Bennett & Tieleman, 2008*). Cyclohexane and octanol measurements for acetamide are labeled *c* and *o* at 0 and 1.3 nm, respectively. Whole-residue Wimley-White measurements of glycine are labeled *w* at 1.3 and 1.7 nm. Cα–Cα: Twice the atomistic PMF for acetamide is shown (black line). *2w* corresponds to twice the Wimley-White value for glycine at 1.3 and 1.7 nm. The water-octanol transfer energy of the Gly dipeptide is labeled d* at 1.3 nm (*Fauchere & Pliska, 1983*). Ala: The atomistic PMF of methane (CH$_4$) is shifted to account for glycine's 1.15 kcal octanol solvation penalty (black). Whole-residue Wimley-White values for alanine are labeled *w*. The PMF of a single CgProt apolar site (ap) is shown in aqua.

**Table 2** **Peptide orientation during 1–2 µs of simulation time.**

| Peptide | Starting state | Helix tilt[a] (°) | $z_{center}$[b] (Å) | $z_{N-term}$[c] (Å) | $z_{C-term}$[c] (Å) |
|---|---|---|---|---|---|
| LS3 | desorbed | 85.1 ± 5.5 | 12.4 ± 1.1 | 11.4 ± 1.7 | 13.4 ± 1.8 |
| LS3 | interfacial | 84.6 ± 5.4 | 12.2 ± 1.2 | 11.0 ± 1.7 | 13.4 ± 1.8 |
| L$_{24}$ | desorbed | 85.4 ± 3.7 | 19.6 ± 1.5 | 19.0 ± 1.5 | 18.4 ± 1.5 |
| L$_{24}$ | transmembrane | 32.5 ± 6.2 | −0.1 ± 0.2 | 16.8 ± 1.8 | −18.2 ± 2.0 |
| (LA)$_{12}$ | desorbed | 88.9 ± 3.8 | 13.6 ± 1.1 | 14.1 ± 1.4 | 13.8 ± 1.5 |
| (LA)$_{12}$ | transmembrane | 35.4 ± 5.4 | 0.7 ± 1.3 | 17.5 ± 1.4 | −17.0 ± 1.7 |
| WALP23 | desorbed | – | – | – | – |
| WALP23 | transmembrane | 41.0 ± 7.3 | 0.9 ± 1.5 | 13.3 ± 1.8 | −10.2 ± 2.5 |

**Notes.**
[a] Mean angle ± standard deviation of helix axis with respect to bilayer normal.
[b] Mean displacement of helix center of mass from bilayer center.
[c] Mean distance of terminal Cα from bilayer center.

single stable orientation within the membrane, with the exception of the desorbed WALP23 simulation, in which the peptide did not adsorb onto the membrane. The kinetic barrier to WALP23 insertion can be explained by the barrier in the PMF for tryptophan at the membrane interface near $z = 2.5$ nm (Fig. 3). The orientation of the helix axis relative to the bilayer normal is shown for each simulation in Table 2, along with the membrane depth of the peptide center of mass. The final equilibrated conformation for each membrane-bound simulation is shown in Fig. 7.
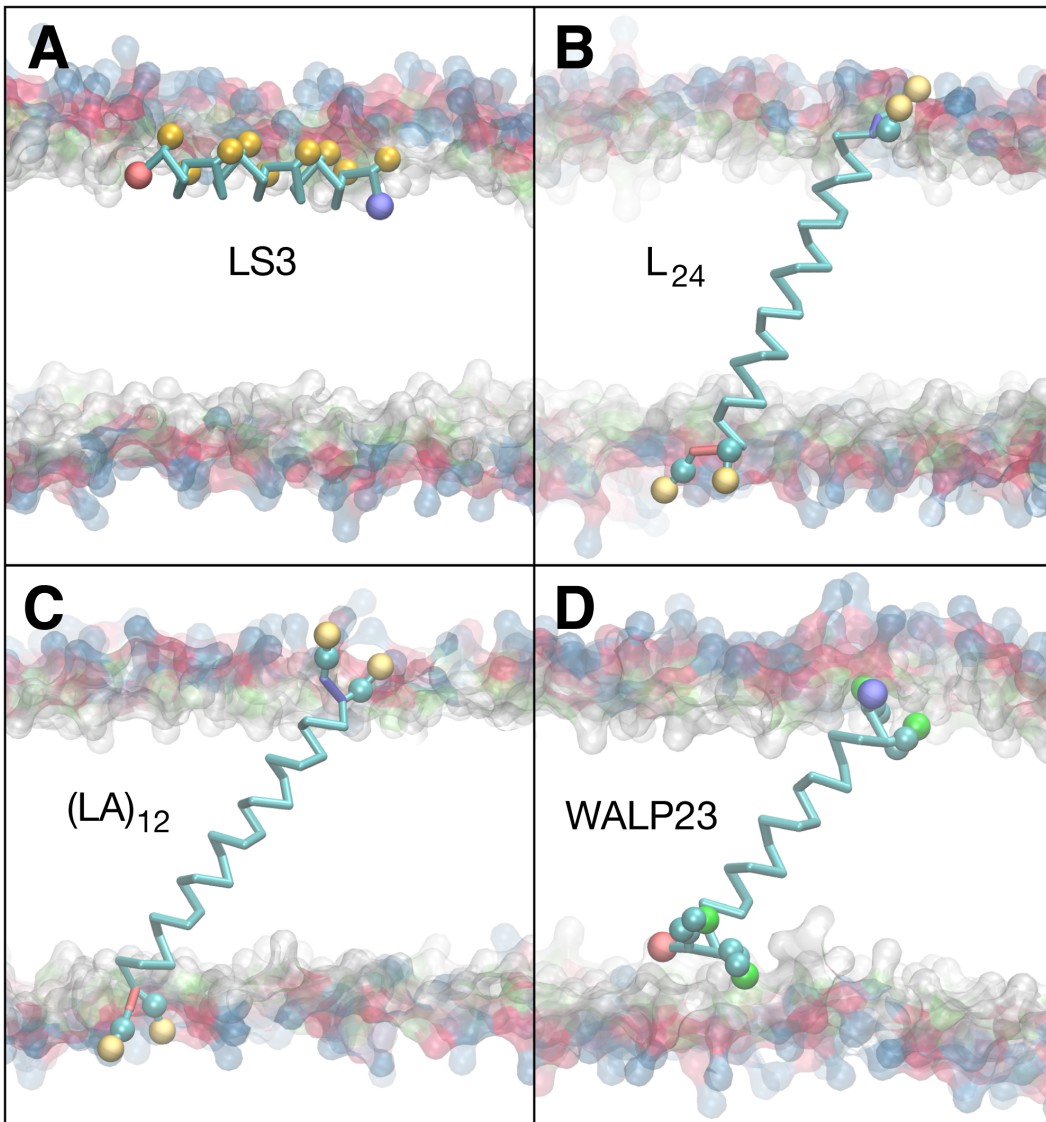

**Figure 7 Equilibrated peptide orientations in μs membrane-bound simulations.** The LS3 peptide inserts in an interfacial orientation (A), while L$_{24}$, (LA)$_{12}$, and WALP23 remain in a transmembrane orientation (B, C & D). The surface of lipid head groups is shown in transparent blue (choline), red (phosphate), green (glycerol) and white (esters). The Cα backbone is traced in cyan with neutral N- and C-termini labeled blue and red, respectively. Spheres are colored for polar sites belonging to serine (gold) and tryptophan (green) sidechains and for the positive charge of lysine (cream).

Interfacial conformations were observed in both unrestrained LS3 simulations that are indistinguishable from each other. The N-terminus sits 2 Å lower in the membrane than the C-terminus (Table 2). Given that the termini were uncharged this is likely a property of the geometry of the peptide. In contrast to CgProt 2.3 simulations with charged termini and the hydrophobic *ca* type backbone in which LS3 preferred the transmembrane configuration (*Fosso-Tande et al., 2017*), the umbrella sampling PMF obtained for insertion of the neutral

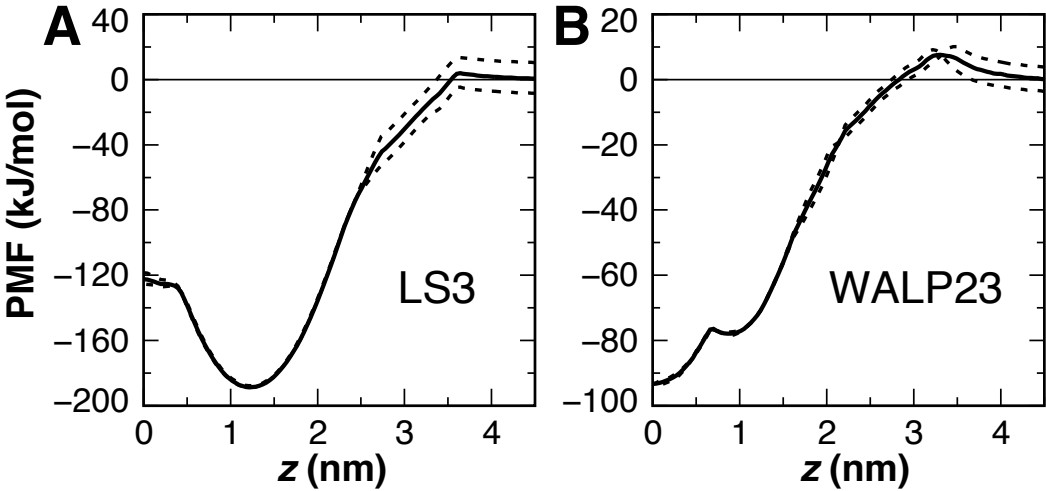

**Figure 8 Peptide insertion PMFs.** Independent PMFs were obtained for the permeation of LS3 (A) and WALP23 (B) in both leaflets of the bilayer (dashed lines). The symmetrized PMF (solid line) reveals a global energy minimum for both interfacial LS3 and transmembrane WALP23.

LS3 confirms that the interfacial conformation is the only preferred energy minimum in CgProt 2.4 (Fig. 8A). The PMF at $z = 0$ is 65 kJ/mol higher in energy than the interfacial state found at a depth of $z = 1.2$ nm. When restrained to the center of the bilayer, LS3 adopts an unstable transmembrane orientation in which the serine sidechains pull lipid head groups into the bilayer interior. While phospholipid flip-flop has not been observed in pure membrane simulations with CgProt, enhancement of lipid flip-flop by transmembrane hydrophilic residues is consistent with recent experiments on endoplasmic reticulum peptide sequences (*Nakao et al., 2016*). The pronounced stabilization of interfacial LS3 is in contrast to unrestrained simulations with the Bond CG model (*Bond, Wee & Sansom, 2008*; *Hall, Chetwynd & Sansom, 2011*), a variant of Martini, in which the interfacial and transmembrane conformations were observed with 80% and 20% frequency, respectively. In situ, the amphipathic LS3 peptide is only expected to adopt a transmembrane orientation when residing in the membrane as part of a homo-oligomeric bundle, which serves as an ion channel (*Nguyen, Liu & Moore, 2013*).

The PMF for the bilayer insertion of WALP23 reveals that the transmembrane configuration centered near $z = 0$ is preferred over the interfacial state in the vicinity of $z = 0.9$ nm by 15.5 kJ/mol (Fig. 8B). The interfacial state is a local energy minimum but has a minimal 1 kJ barrier separating it from the transmembrane state. It is therefore not expected to be stable in unbiased molecular dynamics simulations. The stable insertion of neutral WALP is a considerable improvement in CgProt 2.4. The $24 \pm 5°$ tilt angle obtained by fluorescence spectroscopy (*Holt et al., 2009*) is within two standard deviations of the range observed in the unrestrained simulation ($41 \pm 7°$). In contrast, the hydrophobic *ca* type backbone employed in previous versions causes uncharged WALP peptide termini to submerge in the hydrophobic tail region of the bilayer (Fig. S1). The 93 kJ/mol stabilization of transmembrane WALP in the current version is comparable to the 146 kJ/mol energy

minimum observed for the PLUM force field in a POPC bilayer (*Bereau et al., 2015*). In contrast, transmembrane and interfacial WALP orientations have been observed with 80% and 20% frequency, respectively, in variants of the Martini model (*Bereau & Kremer, 2016*; *Hall, Chetwynd & Sansom, 2011*).

The previous *ca* type hydrophobic backbone enabled the insertion of peptides to be observed within 50 ns of unbiased molecular dynamics simulation. Optimizing the balance of backbone and sidechain energetics in the current version, however, resulted in no membrane insertion events being observed in 8 μs of simulations starting from the desorbed state. This includes the $L_{24}$ and $(LA)_{12}$ peptides, which were observed to adsorb onto the membrane surface but not cross over to the other monolayer on the timescale of the simulation. This can be explained by the presence of a 10 kJ barrier for the polar type backbone sites of non-alanine residues to cross the center of the bilayer (Fig. 6). Additionally, the terminal lysines in the peptides each have a 4 kJ barrier to penetrating the bilayer interface. These studies highlight the need for either manual insertion of a protein into the bilayer or the use of biased sampling methods.

## CONCLUSION

The CgProt nonbond parameters have been refined to improve the permeation behavior for the peptide backbone and amino acid sidechain analogs (*Hu, Sinha & Patel, 2014*). The backbone unit has been made less attractive for the bilayer by assigning the polar site type for non-alanine residues and the apolar site type for alanine. Additional polar sites have been assigned for sidechains Arg, Asp, Lys, Gln, Glu, Tyr, and Phe. The permeation of version 2.4 sidechains correlates well with PMFs from atomistic simulation and experimental hydrophobicity scales based on cyclohexane-to-water and octanol-to-water partition energies. Atomistic simulations with OPLS overestimate the attraction of Gln, Arg and Lys for the ester region of the bilayer. The permeation of Asn and Lys is much improved in CgProt 2.4 compared to v.1, while Gln and Arg experience a favorable attraction similar to atomistic simulations. That Arg is more attractive to the membrane than Lys in CgProt is consistent with the enhanced activity of arginine containing antimicrobial peptides versus their lysine containing counterparts (*Rice & Wereszczynski, 2017*). Note that three sidechain assignments employed in v.2.3 are not recommended based on their permeation behavior: the p.p.pos description of Lys, the ap-p-ap-p description of Trp, and the p-p description of Asn. Previous simulations with v.2.3 revealed that the positive N-terminus has enhanced permeation properties compared to the negative C-terminus (*Fosso-Tande et al., 2017*). The enhanced membrane permeation of positive charges as opposed to negative site types is supported by the CgProt PMF calculations.

Refining the CgProt force field by matching sidechain permeation profiles resulted in improved simulations of membrane-associated peptides. While previous CgProt versions relied on the use of terminal charges to anchor peptides and proteins in the membrane (*Fosso-Tande et al., 2017*), the global energy minimum is now correctly obtained for the interfacial LS3 and transmembrane WALP peptides. The improved backbone and sidechain descriptions in CgProt 2.4 enable the simulation of diverse membrane protein systems

and will serve to guide future model development efforts. The CgProt force field is an effective tool for studies of lipid-protein interactions and protein conformational change. To date protein-protein association has not been studied in CgProt. Refinements in amino acid hydrophobicity are likely key to reproducing experimentally observed dimer structures, which has proved a considerable challenge in CG models such as Martini (*Javanainen, Martinez-Seara & Vattulainen, 2017*). The CgProt tools and parameters needed for implementing molecular dynamics simulations in the Gromacs software package are included in the Supplemental Information, along with equilibrated bilayer coordinates.

## ACKNOWLEDGEMENTS

R.D.H. thanks Jacob Fosso-Tande for valuable assistance.

### Funding
This work was supported by a National Science Foundation grant (MCB-1516826). The funders had no role in study design, data collection and analysis, decision to publish, or preparation of the manuscript.

### Grant Disclosures
The following grant information was disclosed by the author:
National Science Foundation: MCB-1516826.

### Competing Interests
The authors declare there are no competing interests.

### Author Contributions
- Ronald D. Hills, Jr conceived and designed the experiments, performed the experiments, analyzed the data, contributed reagents/materials/analysis tools, wrote the paper, prepared figures and/or tables, reviewed drafts of the paper.

### Supplemental Information
Supplemental information for this article can be found online at http://dx.doi.org/10.7717/peerj.4230#supplemental-information.

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
