# Peer review of "Refining amino acid hydrophobicity for dynamics simulation of membrane proteins"

_PeerJ, doi:10.7717/peerj.4230_

## Round 0.1 · original submission · Major Revisions

Although two of our reviewers are quite satisfied with your paper, I agree with reviewer#3's assesment regarding the need for a much more complete description of the technical parameters of the methodology. I also think that a fuller discussion of the original motivation for this force-field, the changes vs. previous versions and a discussion of its strengths/weaknesses regarding other forcefields would considerably strengthen the appeal of your manuscript/forcefield and be reasonably easy for you to write.

Contra reviewer#3, I do not think Fig.1 requires changes, but I agree with Reviewer#1's request for clearer versions of Figs. 3, 5 and 6.

Your paper would, in my view, benefit from a few additional clarifications:

lines 82-84 "Version 2.4 of the force field developed in the present work assigns [...] the existing polar site (p) type to all other backbone beads" Please state your motivation for this change.

line 90. "The CG temperature of kT = 2.24 kJ/mol is coupled to separate temperature groups for lipid and protein". A few clarifications may be helpful:

a) Why did you write it in kT units instead of Kelvin?
b) Were the temperature and coupling parameters for lipid and protein the same? What was the temperature parameter for solvent?
c) The temperature implied by this value of kT is 269 K. The quoted source states that this value "was chosen for the best behavior across all liquid phase systems studied". It may be helpful for the reader to know exactly how you assessed "best behavior": was it lipid density, diffusion coefficient, etc.? Please clarify.

line 100. Authors state "an additional copy of the sidechain was placed 3.7 nm above the first to ensure converged sampling." I think it might be better to argue (as MacCallum et al. did in their paper) that the second molecule (and its "tethering" to the first one at 37 angstrom) was used to duplicate the number of data collected in a single run and to allow a shorter simulation to span the full movement across the membrane.

lines 136-138 Since you use a CG representations, the exact definition of the theta0 and phi0 angles not immediately obvious. An illustration would be most helpful.

lines 144-147 The 48 windows of the umbrella sampling were generated from a pulling simulation, instead of a continuous simulation. Was that decision taken simply to allow "trivial" parallelization ofthe simulation in different machines, or was it required to allow sufficient sampling and/or decrease auto-correlation artifacts?

lines 236-239 You state:
" Positing that octanol has properties analogous to the ester region of the bilayer, we correlated the octanol solvation energies to the PMF values of MacCallum et al. for transferring solute to a membrane depth of z = 1.3 nm. Excluding outliers Arg and Lys, a good correlation of r = 0.92 is obtained (Table 1). Excluding Asp, Glu and Lys, a correlation of r = 0.78 was found for the PMFs calculated in CgProt 2.4"

It took me a while to understand that Arg and Lys are outliers in McCallum, rather than CgProt. I do not know if other readers might be similarly confused, but I suggest a slight change would make the text clearer:

" Positing that octanol has properties analogous to the ester region of the bilayer, we correlated the octanol solvation energies to the PMF values of MacCallum et al. for transferring solute to a membrane depth of z = 1.3 nm and obtained a good correlation of r = 0.92 upon exclusion of outliers Arg and Lys, (Table 1). For the PMFs calculated in CgProt 2.4, a correlation of r = 0.78 was found upon exclusion Asp, Glu and Lys, "

line 313 You state "In contrast, the hydrophobic ca type backbone employed in previous versions fails to anchor the ends of WALP peptides in the bilayer head group region" Please clarify, as in Fosso-Tande et al. the ends of WALP (charged in both end, hydrophobic ca type) were indeed anchored in opposite leaflets of the membrane. Do you mean to say that WALP, when simulated as neutral in both ends with hydrophobic ca type does not assume the correct conformation? If so, please provide supporting data.

Fig.1 The first bead in the sidechains of Arg, Lys, Gln and Glu (as well as one of the beads of the rings of Tyr/Phe) is a "polar carbon". The rationale behind some of these non-intuitive choices can be found in fig.3. An explicit mention of these unusual choices, however, might be profitably included in the legend to fig 1.

Fig. 3: Why are the experimental transfer energies marked at z=0 at z=1.3 nm? The c,g,o marks clutter the graphs and are very hard to see. I think those energies might be more easily shown on a table or marked in the y-axis.

A comparison of the results from Fig.3 with the results obtained with earlier versions of CgProt would be most helpful in the assesment of the strenghts of the updated parameters.

Reviewer 1 ·

Basic reporting

Figure 3. PMF insertion profiles for mass centers of sidechain analogs relative to the bilayer
center.

The letters (w, g, o, c, etc.) at the plot are very confusing, please use a different mark.


Figure 5. Linear regression, please show a small mark, and then the letter next to the mark, otherwise is hard to understand the data presented here.

Figure 6. CgProt PMFs for one and two-residue backbone moieties. Again, the letters overlap and they are difficult to see. Please add a different mark.

Experimental design

Methods are well described, the results and findings are relevant and interesting.

Validity of the findings

It would be interesting if the authors present how their hydrophobicity force field refinement address the protein-protein interaction in a transmembrane helix dimer.

·

Basic reporting

While the literature cited is failry complete, for the reader to get a complete picture of the field, I suggest to adding the reference to the recent paper by Pawlowski et al who developed a continuous lipid model for coarse-grained simulations of membrane proteins (W Pulawski, M Jamroz, M Kolinski, A Kolinski, S Kmiecik
Journal of chemical information and modeling 56 (11), 2207-2215)

Experimental design

No comment

Validity of the findings

No comment

Additional comments

This is a very good paper that sunstantially advances the development of coarse-grained simulations of membrane proteins. The parameters of the CgProt force field were refined to reproduce the peptide-insertion PMFs calculated by all-atom simulations and experimental hydrophobicities. The resulting force field was shown to perform significantly better regarding the insertion of peptides into lipid membranes.

Reviewer 3 ·

Basic reporting

The manuscript by Hills reports on the refinement of the CgProt 2.4 coarse-grained force field for proteins. This is one of several force fields for protein simulations.

The text and the figures are reasonably clear, with a few exceptions. The graphical quality of Figure 1 is rather poor, I suggest using molecular models instead of chemical formulas.

Literature references are not always appropriate; for example, in the introduction, when writing about other coarse-grained force fields for proteins, the author mentions the MARTINI force field but provides incorrect citations. At line 92, the author refers to himself the idea of temperature coupling to separate heat baths, and to Arnarez the idea of implicit solvent for CG simulations. Bizarre.

The text is far from self-contained, particularly in the methods section:
- the description of the force fields is completely absent; one would need to dig out one or more previous papers to understand the meaning of acronyms for different particle types and the shape of the potential energy function.
- I did not find a description of bonded interactions (except for the restraints on helices).
- How was the position of the CG particles calculated?
- which structural properties of proteins are reproduced? which dynamic properties?
- what is the speedup compared to an atomistic simulation?

Experimental design

The goal of the research is well defined: to correct the flaws present in previous versions of the force field. Based on the results presented, the goal seems to have been achieved.
At the same time, it is not very clear what is the general goal of the force field; what are the pros and cons compared to other force fields available in the literature?

The design of the simulations has no original point. All of the simulations seen here have been done before by others, using other methods and/or force fields.

The description of the methodology is insufficient to reproduce any of the calculations: no mention of the algorithms used for thermostats and barostats, nor the cutoffs (if present).

Validity of the findings

The technical quality is difficult to assess because a description of the force field is missing (see above for details) and the simulation parameters are rather incomplete: no mention of the algorithms used for thermostats and barostats, nor the potential energy functions used, nor the cutoffs (if present).

Methods and figure 3: the AA data has been calculated or extracted from the figures by MacCallum et al.?

Line 218, “The favorable attraction of basic residues for the ester bilayer region has been dubbed snorkeling.” Why would it be valid only for basic residues?
In fact, in a real system, snorkeling does not have much to do with the supposed attraction for the ester group, but rather with the tendency of a charged side chain to remain hydrated. The ester region in a lipid bilayer is generally hydrated, hence the proximity of charged side chains with lipid ester groups. The real physics of the system is hidden by the implicit solvent.

Additional comments

The general philosophy of the force field is not described: what are the target properties? what kind of problems should be tackled by this force field?
Does this force field make progress with respect to other similar, available methodologies?

---

## Round 0.2 · accepted · Accept

The changes have made your paper much more readable and I am happy to accept it for publication in PeerJ.

Reviewer 1 ·

Basic reporting

no comment

Experimental design

no comment

Validity of the findings

no comment

Additional comments

no comment